# Prenatal SAMe Treatment Changes via Epigenetic Mechanism/s USVs in Young Mice and Hippocampal Monoamines Turnover at Adulthood in a Mouse Model of Social Hierarchy and Depression

**DOI:** 10.3390/ijms241310721

**Published:** 2023-06-27

**Authors:** Maria Becker, Denis Gorobets, Elena Shmerkin, Liza Weinstein-Fudim, Albert Pinhasov, Asher Ornoy

**Affiliations:** 1Department of Morphological Sciences and Teratology, Adelson School of Medicine, Ariel University, Ariel 40700, Israel; 2Department of Molecular Biology, Adelson School of Medicine, Ariel University, Ariel 40700, Israel; 3Department of Medical Neurobiology Hebrew, University Hadassah Medical School, Jerusalem 9112102, Israel

**Keywords:** S-adenosyl-methionine, depression, brain monoamine metabolism, USVs

## Abstract

The role of hippocampal monoamines and their related genes in the etiology and pathogenesis of depression-like behavior, particularly in impaired sociability traits and the meaning of changes in USVs emitted by pups, remains unknown. We assessed the effects of prenatal administration of S-adenosyl-methionine (SAMe) in Sub mice that exhibit depressive-like behavior on serotonergic, dopaminergic and noradrenergic metabolism and the activity of related genes in the hippocampus (HPC) in adulthood in comparison to saline-treated control Sub mice. During postnatal days 4 and 8, we recorded and analyzed the stress-induced USVs emitted by the pups and tried to understand how the changes in the USVs’ calls may be related to the changes in the monoamines and the activity of related genes. The recordings of the USVs showed that SAMe induced a reduction in the emitted flat and one-frequency step-up call numbers in PND4 pups, whereas step-down type calls were significantly increased by SAMe in PND8 pups. The reduction in the number of calls induced by SAMe following separation from the mothers implies a reduction in anxiety, which is an additional sign of decreased depressive-like behavior. Prenatal SAMe increased the concentrations of serotonin in the HPC in both male and female mice without any change in the levels of 5HIAA. It also decreased the level of the dopamine metabolite DOPAC in females. There were no changes in the levels of norepinephrine and metabolites. Several changes in the expression of genes associated with monoamine metabolism were also induced by prenatal SAMe. The molecular and biochemical data obtained from the HPC studies are generally in accordance with our previously obtained data from the prefrontal cortex of similarly treated Sub mice on postnatal day 90. The changes in both monoamines and their gene expression observed 2–3 months after SAMe treatment are associated with the previously recorded behavioral improvement and seem to demonstrate that SAMe is effective via an epigenetic mechanism.

## 1. Introduction

Epigenetic changes may underlie the pathogenic mechanism of various diseases such as Autism Spectrum Disorder (ASD) and Major Depressive Disorder (MDD) [1,2]. S-adenosyl-methionine (SAMe) was suggested to be a potent epigenetic drug [3,4]. SAMe, being a principal methyl donor, is involved in many trans-methylation reactions and is also required for the synthesis and degradation of brain monoamine neurotransmitters such as serotonin and dopamine that are related to the pathogenesis of several psychiatric disorders, especially depression [5,6]. A deficit in SAMe concentration was found in the cerebrospinal fluid (CSF) of depressed patients and was normalized by intravenous or oral SAMe administration [7]. SAMe adjunct treatment facilitates and enhances the action of tricyclic antidepressants (TCA) and SSRIs in poor responders with MDD [8,9,10,11].

Functional and molecular alterations in the prefrontal cortex (PFC) and hippocampus (HPC) and monoamine synaptic imbalance have been associated with MDD [6,12,13]. Submissive (Sub) mice serve as an animal model of depression demonstrating depressive-like behavior and are susceptible to stressful stimuli [14,15]. Altered monoamines content in brain areas referred to emotionality and social hierarchy were previously found to be associated with the behavioral phenotype in Sub mice [16,17]. Elevated levels of dopamine and a reduction in norepinephrine levels were measured in Sub mice in a variety of brain areas, whereas 5-HT content was decreased only in the brainstem [17]. These altered monoamine contents in Sub mice brains were associated with impaired locomotion, which apparently reflects the absence of motivation [18,19]. Previously, we demonstrated that treatment of the Dams with SAMe improved depression-like behavior when compared to saline-treated Sub mice but did not affect the early neurodevelopmental milestones in male or female offspring [20]. Some of these improvements were gender related [20]. Prenatal SAMe treatment mainly improved sociability (as tested by the three chambers social interaction test) in either male or female Sub mice. It also increased locomotion (as observed by the open field test) in female mice, but not in males. The evaluation of monoamines concentrations in the prefrontal cortex of SAMe-treated offspring demonstrated significantly elevated levels of serotonin (5HT) and dopamine (DA) and their main metabolites, also in a gender-related manner, whereas no changes were observed in norepinephrine [16]. Prenatal SAMe normalized the expression of several genes related to depression and monoamines synthesis and degradation pathways, also in a gender related manner [16,20]. SAMe also reduced the expression of almost all evaluated genes, mostly in males, but not in females [16].

It was suggested that at early life, changes in brain monoamine content can affect normal pup–mother communications and lead to altered emission of ultrasonic vocalizations (USVs) by the pups [21,22,23,24]. Rodents emit various types of USVs reflecting their sociability at infancy, adolescence and adulthood [25]. These calls reflect not only communication [23,24] but also the emotional state of the animal [24,26,27]. The USVs are important in pup–mother communication in order to achieve parental attention and could also predict the later development of an adult anxiety profile [21]. Changes in the frequency and in the acoustic traits of pups’ USVs may represent a diagnostic and phenotyping tool for various neurodevelopmental and psychiatric disorders, such as autism, anxiety and depression [21]. Mouse USVs are sensitive to changes in brain monoamines levels [22]. A deficit of serotonin, dopamine, vasopressin and oxytocin can reduce or even ablate USV production in pups [22,28]. The evaluation of Sub pups’ UVSs may therefore provide a means of early detection of the effect of prenatal SAMe treatment on the depressive-like phenotype of these mice.

In mice, the frequency range of vocalizations is wide: they can vocalize at 40, 60 or 70 kHz with various frequency modulations, depending on their behavior, sociability or social dominance [29,30] and even at 100 Hz in the case of high social activity [29]. One of the most studied vocalization types in mice are the isolation-induced calls emitted by infant pups during postnatal days 3 to 15–16, with the approximate peak of the emission numbers at day 8 [28,31,32,33,34].

The emission of these ultrasonic vocalizations can be altered by acute or chronic stress conditions [21], social defeat [35], anxiety-like and depressive-like states [36]. In young pups, separation from the mother itself can be an inducer of early-life stress and can increase the emission of USVs [33,37], while nesting smell and mother pheromones can alleviate stress in pups, leading to a reduction in the total number of calls [24,38].

Some of the genetic animal models of depression provide evidence that predisposition to depression can lead to a reduction in the total number of vocalizations in pups [39,40]. It was proposed that genetic resilience to stress is associated with the enhancement of separation induced USVs calls [40].

We previously treated pregnant submissive (Sub) mice exhibiting a depressive-like phenotype with SAMe during days 12–14 of gestation and evaluated the depressive–like behavioral traits in their offspring [16,20]. In the present study, we evaluated the effects of prenatal SAMe on USVs in the first 8 postnatal days in the Sub mouse model of depression and social hierarchy. In addition, we also evaluated the possibility that SAMe could also change, in adulthood, the levels of serotonergic, dopaminergic and noradrenergic metabolism in the hippocampus of Sub offspring prenatally treated with SAMe or saline and the expression of related genes. Using RT-PCR, we also compared the expression levels of some genes involved in the monoaminergic functions and metabolism such as *5ht2a* receptors, *Sert*, *Drd2*, *Tph2*, *Aldh* and *Mao-b.*

## 2. Results

### 2.1. SAMe Effect on Stress-Induced USVs in Sub Offspring

#### 2.1.1. SAMe Effect on Total Number of Emission Calls in Offspring

Prenatal treatment by SAMe significantly reduced the total number of emitted vocalizations in PND4 offspring in comparison to Saline-treated animals (Figure 1). Two-way ANOVA demonstrated a significant effect of SAMe treatment in PND4 offspring (F (1, 117) = 8.102, *p* < 0.001). Bonferroni post-hoc analysis of individual groups revealed that SAMe-treated offspring emitted significantly lower number of calls than Saline-treated offspring: 179.12 ± 21.2 (n = 32) vs. 340.7 ± 45.1 (n = 21), respectively, (*p* = 0.001). Representative sonograms modulated to the hearing diapazone of the human ear are in Appendix A. However, this treatment effect disappeared on PND8. The total number of emitted vocalizations in PND8 offspring were equal between groups (Figure 1). Despite the fact that the mean number of the vocalizations was different at PND4, their emission intensity was not altered.

#### 2.1.2. SAMe Effect on the Distribution of Call Types

To investigate which type of calls contribute to the variation in the call numbers between the groups and/or during early postnatal development, we used the VocalMat machine learning algorithm [41], allowing us to evaluate the bioacoustic compositions of the calls. We discovered the following tendencies in the calls’ composition distribution at postnatal days 4 and 8. The predominant type of calls at PND8 and within different treatments were flat calls (Figure 2C,D). At the PND4, a repertoire of the calls was different between the groups. Within Saline-treated offspring a flat as well as one frequency step up calls were the most abundant (Figure 2A). However, in SAMe group, upward frequency modulated and short calls were met more often than others (Figure 2B).

Within both Saline- and SAMe-treated offspring during maturation of the pups from postnatal day 4 to 8, we found an increase in the number of one frequency step down calls, in the two-frequency step calls as well as in the upward frequency modulated syllables. In addition, we observed a reduction in the number of flat and one frequency step up calls, with no differences between the groups (Figure 2C,D).

The most frequently emitted call types at postnatal day 4 in the offspring were flat, one frequency step up and one frequency step down. We therefore analyzed the SAMe effect on the quantitative emission of these specific calls (Figure 3). This analysis revealed that SAMe treatment affected the number of mentioned call types at PND4 and PND8 (Figure 3). Three-way ANOVA analysis revealed a significant effect of SAMe treatment (F (1, 65) = 4.89, *p* = 0.01) on the number of syllabi. Multiple comparison Bonferroni’s analysis demonstrated a significant reduction in the number of emitted flat (*p* = 0.008) and one frequency step up (*p* = 0.04) call numbers by SAMe at PND4 compared to controls. On PND8, SAMe and control offspring emitted almost an equal number of both flat and frequency step up calls in their recorded sonograms (Figure 3A,C). However, one frequency step down calls had revealed an opposite tendency, and their mean number had increased in the SAMe-treated offspring at PND8 (Figure 3B).

The analyses of acoustic characteristics of calls as Duration, Frequency and Intensity of specific call types did not reveal any differences among the treatment groups at PND4 or PND8.

### 2.2. SAMe Effect on Serotonin Metabolism in the Hippocampus

The hippocampal tissue levels of 5HT and its primary end-product, 5-HIAA, and the ratio of 5-HIAA to 5HT were estimated in the HPC. The 5-HT undergoes enzymatic degradation by monoamine oxidase A (MAO_A_) and an aldehyde dehydrogenase, resulting in the production of the end-metabolite 5-Hydroxyindoleacetic acid (5-HIAA). The ratio of 5-HIAA to 5HT serves as an estimated index of the changes in the rate of release of 5HT into the synapse [42,43,44].

SAMe increased the levels of 5HT in the HPC compared to control mice in both male and female Sub mice (Figure 4A). Two-way ANOVA revealed a significant effect of SAMe treatment (F (1, 6) = 12.9), but not of gender, *p* < 0.015 in Sub mice. Paired *t*-test analysis of individual groups showed similar effects of SAMe treatment in both male and female HPC (*p* < 0.03) in comparison to saline controls. However, the levels of 5HIAA and of the 5-HIAA/5-HT ratio in hippocampi were unchanged (Figure 4B,C). These results indicated that 5HT synthesis was increased in Sub mice hippocampi by SAMe treatment without any changes in 5HT degradation pathways.

### 2.3. SAMe Effect on Dopamine, Dopamine Metabolites and of Metabolite/Dopamine Ratio in the Hippocampus

Dopamine levels in the hippocampi of Sub mice were unchanged following SAMe administration (Figure 5A). The metabolic DA degradation, initially by MAO, followed by ALDH, leads to DOPAC formation. DOPAC is degraded by Catechol-O-methyl transferase (COMT), producing Homovanillic acid (HVA). We therefore estimated DOPAC and HVA levels. For the estimation of dopaminergic activity, we calculated the ratio of DOPAC to DA, the ratio of DOPAC to HVA and of DOPAC + HVA to DA.

SAMe facilitated DOPAC formation in a sex-related manner. Two-way ANOVA did not reveal any significant effects of SAMe treatment or gender in the HPC DOPAC level of Sub mice compared to controls. However, multiple pair comparison *t*-tests demonstrated a higher concentration of DOPAC in females (*p* < 0.03), but not in males (*p* = 0.6) (Figure 5B). However, the concentration of HVA (Figure 5D) and metabolite ratios remained unchanged in both sexes (Figure 5C–F).

### 2.4. SAMe Effect on Norepinephrine Metabolism in the Hippocampus

The activity of norepinephrine neurons was estimated by measurements of the HPC levels of NE, its major metabolite 4-hydroxy-3-Methoxyphenylglycol (MHPG) and by the ratio of MHPG to NE. MHPG is a major metabolic product of enzymatic degradation of NE involving COMT, MAO-A and aldehyde dehydrogenase enzymes (reviewed in [45]). Analysis of the NE and MHPG levels, and the MHPG to NE ratio did not reveal any effect of SAMe treatment in comparison to Saline-treated mice (Figure 6A–C).

### 2.5. SAMe Effects on Tph2, Mao-b and Aldh Gene Expression in the in Hippocampus

We measured the expression level of *Tph2,* a gene coding the enzyme that is involved in serotonin formation from tryptophan. In addition, we measured the expression levels of monoamine oxidases genes B (*Mao-b*) and the *Aldh* gene that are involved in the dopamine and serotonin degradation pathways.

Two-way ANOVA analysis and Bonferroni multiple pair comparisons did not reveal significant differences in the expression of the *Tph2* gene, *Aldh* gene and *Mao-b* in the Sub mice hippocampus (Figure 7A–C).

### 2.6. SAMe Effects on Htr2a, Sert, Drd2 Receptor Genes Expression in the Hippocampus of Sub Mice

Two-way ANOVA analysis of the expression levels of the *Htr2a* receptor gene determined the effect of SAMe treatment (F (1, 6) = 20.5; *p* = 0.022). Multiple paired *t*-tests demonstrated a prominent reduction in the expression of the *Htra2* gene (*p* = 0.006) in both male and female Sub hippocampi in comparison to controls (Figure 8A). A Bonferroni multiple pair comparison demonstrated significant differences in the 2^−DDCt^ of expressed genes in males (*p* = 0.002, n = 6) and in females (*p* = 0.03, n = 6) in comparison to Saline control (Figure 8A).

A Bonferroni multiple pair comparison analysis of serotonin transporter (*Sert*) gene expression exhibited significant differences only in male (*p* = 0.03, n = 6) hippocampi treated by SAMe compared to controls (Figure 8B).

The most prominent effects of SAMe were observed on the expression levels of the *Drd2* gene in female hippocampus. Two-way ANOVA analysis demonstrated a significant effect of SAMe treatment (F (1, 6) = 9.989, *p* = 0.025) and the effect of sex (F (1, 6) = 11.45, *p* = 0.02). A Bonferroni multiple pair comparison analysis defined the increase in 2^−DDCt^ measured for the expression of *Drd1* only in females treated with SAMe in comparison to controls (*p* = 0.02) (Figure 8C).

## 3. Discussion

The role of hippocampal monoamines and their associated genes in the etiology of depressive-like behavior in the Sub strain of mice and the meaning of changes in USVs emitted by these pups remains unknown. In previous studies, we evaluated the effect of prenatal SAMe treatment on the behavioral depression-like phenotype of SAMe mice in adulthood [16,20]. SAMe alleviated sociability aversion and reduced changes in the expression of genes related to depression and to monoamines metabolism in the prefrontal cortex (PFC) of Sub mice. In the current study, we aimed to elucidate the effects of SAMe monoamine levels and the expression of several genes related to monoamine metabolism such as *Tph2*, *Aldh* and *Mao-b* and in genes related to synaptic transmission, specifically *Htr2a*, *Sert* and *Drd2*.

In addition, we studied the effects of prenatal SAMe on the isolation-induced USVs calls emitted by the pups following separation from their mothers.

We found that SAMe induced a reduction in the number of emitted calls by the pups on day 4 but not on day 8. SAMe also induced changes in the concentrations of dopamine and several metabolites, serotonin and several metabolites, but not in epinephrine, norepinephrine and metabolites.

### 3.1. SAMe Effect on USVs Emitted by Stressed Pups in Relation to Depression and Sociability

We found a reduced total number of emitted vocalizations only in PND4 offspring in comparison to Saline-treated pups. Whereas at the PND8, which is known as the day at which pups emit the highest rate of calls [28,31,32,33], either SAMe or Saline-treated pups emitted a similar number of calls. The most frequently emitted calls that were reduced by SAMe were flat, one frequency step-up, short and step-down calls. Other acoustic characteristics of emitted by pups’ calls, such as duration, frequency and intensity of specific call types, remained unchanged by SAMe treatment either at PND4 or PND8.

Various pathological conditions in rodents may induce USVs alterations in the pups. An overall reduction in the number of calls were found in diverse neurodevelopmental disease models [31,34,46]. A reduction in the number of emitted calls or a decrease in the variety of calls [47], as well as abnormal changes in acoustic characteristics such as duration [48] were reported in mouse models of autism. A total reduction in call numbers, intensity and duration were also reported in a mouse model of Parkinson disease [49].

The specific meaning of call types and their relation to pup–mother communications is still unknown. We may presume that the SAMe-induced reduction in emitted calls on day 4, with changes in emission of flat, one frequency step up, short and step- down call types, is related to pups’ anxiety levels, induced by isolation from the mother. We previously reported the absence of a SAMe effect on anxiety in the adult 90 days old offspring, as measured by an elevated plus maze test [20]. However, at adulthood, SAMe-treated offspring demonstrated increased locomotion in the open field and improved sociability in a three-chamber test [20]. As was suggested, pups’ communication with their mother via USVs is primarily linked to social behavior [21,50]. We therefore assume that the changes induced by SAMe treatment in the number of calls emitted by pups at PND4 may be associated with higher motivation, reflected by improved sociability and increased locomotion in adulthood [18,19].

In the current study, we also tried to elucidate how hippocampal monoamines metabolism interplay with the altered ultrasonic vocalization emission. In pharmacological studies carried out by Cagiano et al. [51] and by Cuomo et al., [52] they found that DRD2 receptor blockers, but not DRD1, can reduce USV vocalizations in rat pups. This was endorsed by a genetic model where DR2-KO rat pups emitted reduced total vocalizations [53]. However, Wang et al. [54] demonstrated that D2 receptor KO did not lead to any alternations in total vocalization numbers or composition. It is unclear how the SAMe-induced elevated expression of the *Drd2* gene only in females, observed in the hippocampi at adulthood, is related to the reduced number of emitted USVs in the SAMe-treated pups.

Another clue to the explanation of the reduced USVs emission may be related to the changes in serotonin metabolism. We found an elevated level of serotonin in the hippocampus of SAMe-treated mice and reduced expression of *Htr2a* receptor in both sexes. However, 5-HT receptor knock-out female mice emitted higher amounts of USVs at PND8 [55]. In our current study, the reduced number of emitted calls in pups by SAMe was found only at PND4 and was accompanied by a reduced expression level of the *Htr2a* gene in both sexes and *Sert* gene expression elevation only in males. Another research group underlined a role of *Tph2* in early-age abnormalities of the ultrasonic isolation calls where knock-out of the *Tph2* pups emitted a reduced number of USV calls on postnatal days 3 and 6, while the acoustic characteristics of such vocalizations were unaltered [56].

### 3.2. Hippocampal Monoamines in Relation to Previously Described Changes in the Prefrontal Cortex

Brain monoamines metabolism and turnover represent an ongoing equilibrium between synthesis, storage and catabolic degradation that reflects neurotransmitter release in response to nerve impulse activity [12]. Hence, we evaluated the levels of serotonin, dopamine, norepinephrine and their main end metabolites.

In previous studies, we performed similar studies on SAMe-treated and control Sub mice and on day 90, we evaluated the changes induced in the prefrontal cortex (PFC) as well as behavioral studies [16]. We therefore compared the hippocampal content of monoamines and the expression of genes with the findings observed in the PFC that we reported previously [16]. These biochemical and molecular data in the PFC and the HPC comparisons are presented in Table 1 and Table 2.

Many, but not all, biochemical and molecular changes in the PFC and HPC were similar in both studies. The few differences observed (i.e., in DA levels and in the expression of Sert, *Mao-b* and *Drd2* genes) can be attributed to the fact that these are different regions of the brain with different roles in the pathogenesis of depressive-like behavior.

SAMe increased the levels of 5HT in the hippocampus compared to control mice in both male and female Sub mice. However, the levels of 5HIAA metabolite and the 5-HIAA/5-HT ratio in the hippocampi were unchanged. SAMe increased the levels of 5HT in Sub mice PFC only in females (Table 1). We assume that these findings may reflect two independent metabolic processes and the SAMe effects on the hippocampus are more pronounced than on the PFC, in line with previous findings that depressive-like behavior in the Sub mice involves profound changes in proteins related to a variety of hippocampal nervous functions [57,58]. However, the expression of the *Tph2* gene that codes for the TPH2 enzyme, which is involved in Serotonin synthesis from tryptophan [26], was not changed by SAMe treatment in either the hippocampus or PFC.

A possible explanation is that SAMe leads to a slowed serotonin degradation pathway. This second assumption is in line with the study that demonstrated that the inhibition of serotonin degradation by acute paroxetine treatment (i.p, 3 mg/kg) resulted in elevation of 5HT levels in the PFC of Sub mice, and decreased both the level of 5-HIAA and the ratio of 5HT to 5-HIAA [17]. Hence, it is reasonable to assume that SAMe inhibited 5HT degradation, similarly to that of Paroxetine. However, the *Htr2a* was significantly decreased by SAMe in both male and female hippocampi, while *Sert* gene expression was increased only in males.

Dopamine (DA) levels in the hippocampi of Sub mice were unchanged following SAMe administration, while they were increased in both sexes in the PFC. The metabolic degradation of dopamine, initially by MAO, followed by ALDH, leads to DOPAC formation. SAMe facilitated DOPAC formation only in female’s hippocampi. However, the ratio of the main metabolites to dopamine concentrations remained the same.

DA is involved in the regulation of motor activity, emotions, motivation and reward, and has been associated with cognition [59]. It was previously found that Sub mice have reduced locomotion, which may reflect the absence of motivation [16,18,19]—all processes that are under dopaminergic system regulation. In contrast to the findings in the PFC, dopamine levels in the hippocampi were not altered by SAMe treatment, and DOPAC levels were elevated only in females. This again emphasizes the differences between the PFC and HPC in our animal model of depressive-like behavior. It seems that the increased sociability index and increased locomotion in the open field test previously reported by us as being induced by SAMe in Sub mice are associated with elevated levels of DA and their metabolite DOPAC in the PFC but not in the hippocampus [16]. This is also manifested by the lack of changes in the expression of genes involved in DA and NE degradation pathways, *Aldh* and *Mao-b*. In the study by Murlanova et al., acute treatment with paroxetine lead to decreased levels of dopamine in the brain areas of Sub mice, but no alterations in the levels of dopamine metabolites, DOPAC and HVA were reported [17].

The brain is very susceptible to the damage of oxidative stress as well as nitrosative stress. Calabrese et al. have shown that physiological concentrations of NO are involved in a variety of regulatory mechanisms in the brain, while high doses are neurotoxic [60]. Generally, the endogenous protective mechanism known as cellular stress response might defend against stressful conditions and activate the pro-survival pathways. The cellular stress response is regulated by vitagenes, protective genes endorsing the production of antioxidants and antiapoptotic molecules such as the heat shock protein (Hsp) family, including heme oxygenase-1 (HO-1), Hsp72, Thioredoxin/Thioredoxin reductase and Sirtuins [61,62], which can be transcriptionally upregulated by transcription factor Nrf2 (Nuclear erythroid 2- related factor 2) (reviewed in [2,4]).

SAMe, in addition to its role in DNA methylation, may also serve as an important antioxidant [3]. Hence, it is possible that in our study, SAMe normalized some of the biochemical and molecular changes observed in the untreated Sub mice via its antioxidant capacity. Indeed, in our previous studies on a mouse model of Autistic-like behavior induced by early postnatal treatment with VPA, we found that concomitant treatment with SAMe alleviated not only the autistic-like behavior but also the oxidative stress induced by VPA, as observed in the prefrontal cortex at 60 days of age [63].

In addition, in a rodent model of galactose-induced brain aging, Zhang et al. [64] showed that chronic treatment with SAMe enhanced antioxidant capacity and reduced the expression levels of α7 nicotinic acetylcholine receptor (α7nAChR), nuclear factor erythrocyte 2-related factor 2 (Nrf2) and heme oxygenase 1 (HO-1) genes in the hippocampus, and improved cognitive impairment in rat behavior [64].

It was shown that rapid degradation of 5-HT, NE and DA by MAO enzyme leads to the formation of potentially neurotoxic species, such as hydrogen peroxide and ammonia, that resulted in the production of reactive oxygen species (ROS), inducing mitochondrial damage and neuronal apoptosis (reviewed in [65]). We assume that SAMe inhibited the degradation of monoamines which, in turn, may diminish the formation of ROS and NOS. However, in our present study, we did not measure the redox potential to elucidate the possible effect of SAMe on redox homeostasis.

## 4. Materials and Methods

### 4.1. Animals

All animal experiments were approved by the Institutional Animal Care and Use Committee of Ariel University, Israel, and institutional guidelines for the proper and humane use of animals in research were followed (approval # IL-188-12-19). The populations of submissive (Sub) mice were raised from the outbred Sabra strain (Envigo, Ness Ziona, Israel), selectively bred for 48 generations based on the food competition Dominance Submissive Relationship (DSR) paradigm [15,66,67].

Sub female mice were mated with Sub males and monitored at early morning for the presence of vaginal plug, which is considered as gestational day (GD) 0.

Treatment: S-adenosyl-methionine (SAMe) was kindly provided as a gift by Prof. Szyf M., (McGill University, Faculty of Medicine). During days 12–14 of gestation, Submissive females were treated daily by oral gavage with 20 mg/kg of SAMe or by Normal Saline (controls). Pregnant females were housed individually in cages in a colony room (12:12 L:D cycle with lights on 07:00–19:00 h, 23 ± 2 °C, ambient humidity) and provided with standard laboratory chow and water, ad libitum. Further, their offspring were separated at weaning according to gender (≈day 21) in groups of five per cage (see chart flow in Figure 9).

### 4.2. USV Recording and Analysis

The emitted USVs were measured at postnatal days 4 and 8. The pups were separated from their mothers for 30 min in individual chambers, and after 30 min of isolation, were subjected to USV recordings. Each pup was placed in a plastic rectangle chamber (11 × 6 cm, height of the walls 5 cm) located in a sound-insulating Styrofoam box for 3 min for recording [20]. USV recordings were performed using the Ultravox 2.0 microphone (Noldus Information Technology, Wageningen, The Netherlands) placed 10 cm above the pup in the Styrofoam box, at the sample rate of 250 kHz. After the recording, pups were returned to the home cage with the mother. The acquired data was filtered using lower cut-off 20 kHz filter to diminish the effects of the background noise. After the data were collected, a filtered file was exported in WAV format by the software (Ultravox (2.0)). Then, registered sonograms from 32 SAMe were treated and 21 Saline-treated pups were evaluated using a machine-learning algorithm [41], and the total number of calls, call types and their acoustic parameters were calculated.

There were multiple attempts to classify the repertoire of ultrasonic vocalization calls or isolation syllables in mice with the various nomenclatures. In our study, we adopted the classification described by Grimsley et al. [31], modified by Fonseca et al. [41].

The following call types were detected by the algorithm:Short syllables—short vocalizations with durations shorter than 5 ms.Flat syllables—calls with small frequency modulations or without frequency modulations (less than 5 kHz).Upward syllables—the frequency modulation at the end of the call is ≥6 kHz than the frequency at the start.Downward syllables—the frequency modulation at the end of the call is ≤6 kHz than the frequency at the start.Complex—single syllables consisting of combined bidirectional frequency modulations greater than 6 kHz.One frequency step up—two-component vocalizations with no time difference between them and a second component where the emission frequency is higher than the emission frequency of the first by ≥10 kHz.One frequency step down—two-component vocalizations with no time difference between them and a second component where the emission frequency is lower than the emission frequency of the first by ≥10 kHz.Chevron syllables—a type of call with a reversed U shape, in which the difference between the highest point and the start and end points is ≥6 kHz.Reverse chevron—a type of call with a U-shaped syllable type, in which the difference between the lowest point and the start and end points is ≥6 kHz.Multiple frequency steps—a more than three-component vocalization, with differences between the frequencies of each component’s emission.Two frequency steps—a three-component vocalization with no time delay between the emission of each component, where second component differs from the first by ≥10 kHz, and the third one differs from the second by ≥10 kHz.

Statistical analysis was performed using platform RStudio 4.2.1 (Posit, Boston, MA, USA, RStudio Team http://www.rstudio.com/, accessed on 1 May 2023) utilizing the following packages: Seewave [68], tuneR [69], ggplot2 [70], viridis [71], grid [72] and gridExtra [73] and custom-written macros. Spectogram plotting was performed using Audacity^®^ sound processing software (Version 3.3.3) [74]. An oscillogram plotting was performed using MATLAB and Signal Analyzer Toolbox release R2023a (The MathWorks, Inc., Natick, MA, USA). Processed files were subjected to custom-written macros for R to extract the numerical values of the characteristics of interest.

### 4.3. Monoamine Measurements

At the end of the behavioral evaluations on day 90, mice were killed by exposure to CO2 and their hippocampal formation (HPC) was collected, frozen in liquid nitrogen and stored at −80 °C for the study of monoamines and gene expression. Sub offspring treated with SAMe were compared to Saline-treated Sub mice offspring. Dopamine (DA) and its metabolites, 3,4-dihydroxyphenylacetic acid (DOPAC), homovanillic acid (HVA); serotonin, a 5-hydroxytriptamine (5-HT) and its metabolite, 5-hydroxyindoleacetic acid (5-HIAA); norepinephrine (NE) and its metabolite, 3-methoxy-4-hydroxyphenylglycol (MHPG) contents were measured in the brain by using an HPLC ALEXYS neurotransmitters analyzer with column acquity UPLC BEH C18, 1.7 µm, 1 × 100 mm (Waters) and electrochemical detector (ECD). For the analyses of brain monoamines and their metabolites (5-HT, 5-HIAA, NE, MHPG, DA, DOPAC and HVA), the frozen PFC was prepared according to the ALEXYS™ neurotransmitter analyzer protocol.

Briefly, the tissue samples were homogenized in 0.2 M perchloric acid (0.5 mL/100 mg wet tissue), left in ice for 30 min and centrifuged at 20,000× *g* for 15 min at +4 °C. The supernatants were filtered (PhenexTM Nylon, 0.45 μm) and adjusted to pH 3.0 with 1 M sodium acetate.

For analysis, 2 μL of each sample were injected into the HPLC-ECD system at a temperature of 37 °C with a flow rate of 50 µL/min and a backpressure of about 250 bar. The mobile phase included 100 mM phosphoric acid, 100 mM citric acid, 0.1 mM EDTA-Na2, 6.9 mL 50% NaOH, 600 mg/L octanesulfonic acid sodium salt and 8% acetonitrile.

The quantity of monoamines in a total volume of 20 μL in each sample was measured by DataApex chromatography software (version 7.0, Prague, The Czech Republic) and final concentrations expressed as ng/g of wet brain tissue were adjusted to commercially obtained standards: 5-hydroxytryptamine (5-HT, Supelco, St. Louis, MO, USA, CAS # 50-67-9), 5-5-hydroxyindoleacetic acid (5-HIAA, Merck, Rahway, NJ, USA, CAS #: 54-16-0), dopamine (DA, Merck, CAS #: 62-31-7), 3,4-Dihydroxyphenylacetic acid (DOPAC, Merck, CAS#: 102-32-9), norepinephrine (NE, Merck, CAS#: 51-41-2), 4-hydroxy-3-Methoxyphenylglycol (MHPG) piperazine (Merck, CAS #: 67423-45-4), homovanillic acid (HVA, Supelco, CAS #:306-08-1). For 5-HT and 5-HIAA, stock standard solutions were prepared in 0.1 M acetic acid including 1 mg/mL EDTA-2Na. All other standards were prepared in 0.1 N HCl including 1 mg/mL EDTA-2Na. Standard solutions were serially diluted in 0.02 M acetic acid including 10 μM EDTA-2Na to prepare working concentrations of 250 to 0.39 pg/μL required for the calibration.

We measured the levels in the HPC of 5-HT, 5-HIAA and 5-HT turnover (5-HIAA/5-HT) and used 5-HIAA as an indicator of MAO-A activity [75]. The ratio of DOPAC to dopamine indicates the rate of dopamine metabolism, whereas changes in the levels of dopamine metabolites, DOPAC and HVA, reflect changes in MAO activity. Dopaminergic neuronal activity can be further estimated by calculation of DO-PAC + HVA/dopamine ratio. This ratio, which indicates alterations in the rate of dopamine turnover, was found to be changed in a number of pathological conditions (experimental model of Parkinson’s disease or following stroke) as well as during drug treatments [76,77].

### 4.4. Gene Expression Studies

The collected hippocampi were inserted in to 300 µL of DNA/RNA Shield (#Cat: ZR-R1100-50, Zymo Re-search, Irvine, CA, USA), frozen by liquid nitrogen and stored at −80 °C.

In the hippocampi, we studied by RT-PCR the expression of genes related to serotonin synthesis degradation: *Tph2*, *Mao-b* and *Adlh*, serotonin receptor *Htr2a* and Serotonin Transporter *Sert*. In addition, we studied the expression of genes related to dopamine enzymatic degradation: *Mao-b*, and dopamine receptor *Drd2* gene. The analysis was normalized to the housekeeping gene *Gapdh*. mRNAs were purified with Direct-zol™ RNA MiniPrepPlus (Cat#: ZR-R2081, Zymo Research, Irvine, CA, USA) and quantified according to absorbance at 260 nm using spectrophotometry. Complementary DNA (cDNA) was transcribed from 1 mg RNA using qScript cDNA Synthesis Kit (Cat#:95047-100-2, Quantabio, Beverly, MA, USA). cDNA samples were analyzed for the expression of the genes using commercially synthesized primers (Merck, Israel, presented in Table 3). Amplification was performed using PerfeCTa SYBR Green FastMix (Cat #: 95074-012; Quantabio, Beverly, MA, USA) using the following conditions: 30 s denaturation at 95 °C, followed by 30 s annealing at 60 °C for a total of 40 cycles in Agilent AriaMx qPCR thermal cycler. The obtained cycle thresholds (Ct) values for gene of interest (a target sample) and housekeeping gene (GAPDH) for both the treated and untreated samples were used to calculate the relative fold gene expression level 2^−∆∆CT^ [78]. Gene symbols are written in italicized letters with only the first letter in upper-case. Protein symbols are written in regular, upper-case letters.

### 4.5. Statistical Analysis

For statistical significance of monoamines content in the HPC in relation to gender, we performed a two-way analysis of variance (two-way ANOVA), followed by post hoc Bonferroni test or multiple paired *t*-test comparison analysis. For all analyses, adjusted *p* ≤ 0.05 was considered significant. GraphPad Prism 9.0 software was used for graph preparation and statistical analysis.

## Figures and Tables

**Figure 1 ijms-24-10721-f001:**
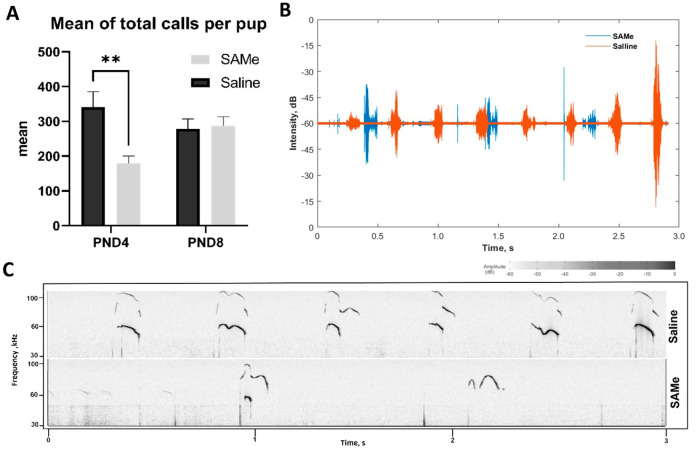
The mean of call numbers differs between the groups. (**A**)—bar charts representing the average number of calls emitted by pups on postnatal days 4 and 8, respectively. Bars reflect mean values, error bars represent SEM. Two-Way ANOVA, **—*p*-value of multiple comparisons < 0.01. (**B**)—representative oscillogram of a three-second sample of the USV recording. USV oscillogram of SAMe-treated offspring (blue trace) overlaid with that of Saline-treated offspring (orange trace), reflecting similarity in the emission intensity between groups reflecting a higher number of the calls during a certain time. (**C**)—representative sonograms obtained from sample recording of a postnatal day 4 SAMe- (upper panel) or Saline (lower panel)-treated offspring, reflecting differences in emission frequency in the treatment groups, with the SAMe-treated pups emitting vocalizations much more rarely than the Saline-treated ones.

**Figure 2 ijms-24-10721-f002:**
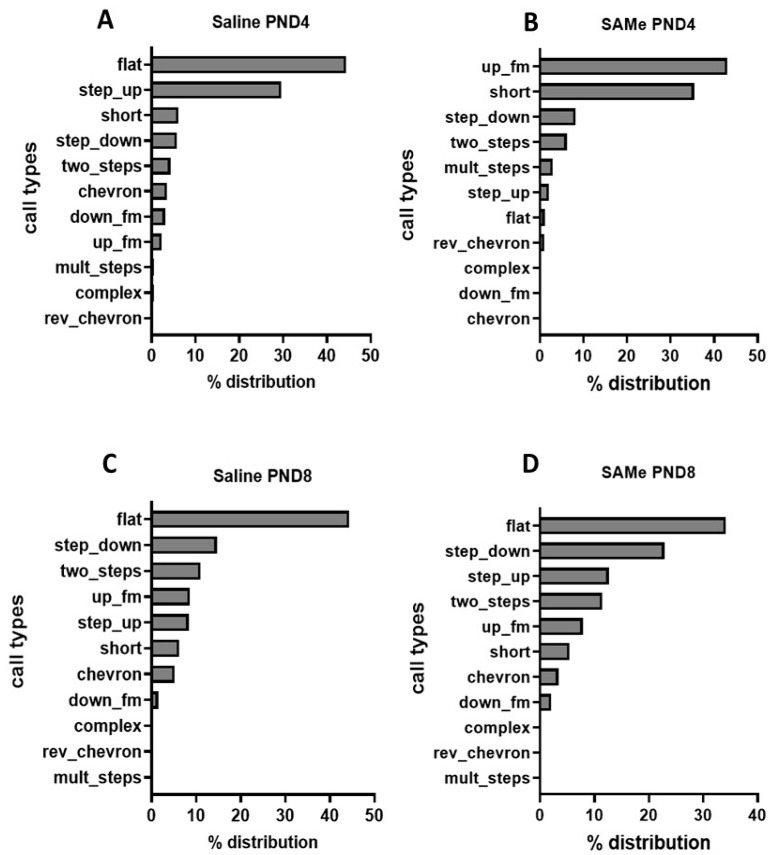
Call type distribution between experimental groups. The charts reflect the percentage distribution of the different call types within the treatment groups and ages, sorted in a decreasing order. (**A**)—Saline-treated offspring’s call distribution at postnatal day 4. (**B**)—SAMe-treated pups’ call distribution at PND4. (**C**)—Saline-treated pups’ calls at PND8. (**D**)—distribution of the calls within the SAMe group at PND8.

**Figure 3 ijms-24-10721-f003:**
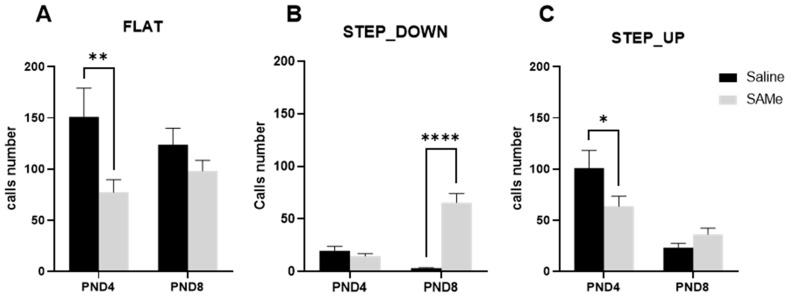
Prenatal treatment with SAMe leads to the alternation of the specific syllables types. SAMe given prenatally, changed the number of flat, frequency step down and step up calls emitted by pups at PND4: (**A**)—The number of emitted flat calls were significantly lowered by SAMe, compared to the control group (n = 21) at PND4, but not on day 8; (**B**)—SAMe increased the number of emitted one frequency step down calls at PND8. (**C**)—The number of frequency step up call emissions was reduced by SAMe at PND4 compared to control. Bar charts represent mean of call numbers, error bars represent SEM; Two-Way ANOVA, *—*p*-value of multiple comparisons < 0.01, **—*p*-value < 0.001, ****—*p*-value < 0.0001, n = 32 pups for SAMe and 21 for Saline group, respectively.

**Figure 4 ijms-24-10721-f004:**
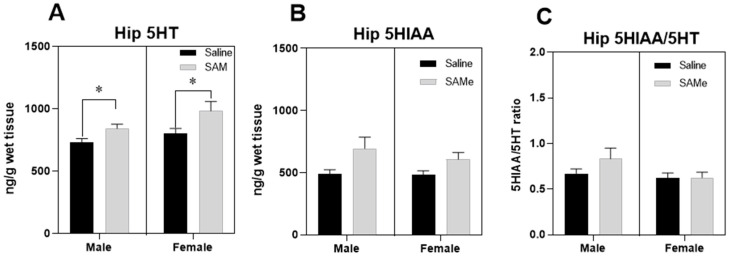
SAMe effect on Serotonin turnover in the Hippocampus. SAMe treatment significantly elevated the levels of serotonin in the HPC of both male and female mice (**A**). The levels of HIAA (**B**) and of the 5-HIAA/5-HT ratio (**C**) remained unchanged in both sexes. The data are presented as mean ± SEM. Levels of 5-HT and 5-HIAA are presented in ng/mg, n = 6 for each group. * *p* < 0.05 compared to control mice of the same gender.

**Figure 5 ijms-24-10721-f005:**
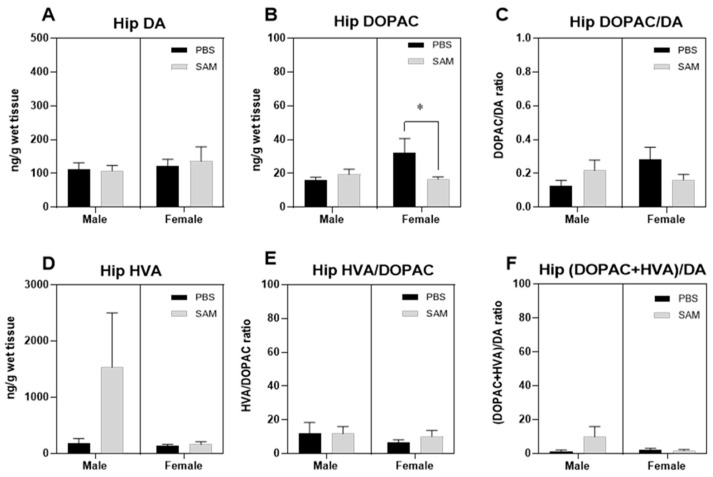
SAMe effect on dopamine metabolism (**A**). The levels of dopamine in the hippocampus of Sub mice were unchanged by SAMe treatment (**B**). The levels of DA catabolic product, DOPAC, were higher in the hippocampus of female Sub mice; however, the differences in the ratio of DOPAC/DA did not reach statistical significance (**C**). The levels of HVA (**D**), HVA/DOPAC index (**E**), and overall turnover of DA were unchanged (**F**). The data are presented as mean ± SEM. The levels of DA, DOPAC and HVA are presented in ng/mg, n = 6 for each group. * *p* < 0.05 compared to control mice of the same gender.

**Figure 6 ijms-24-10721-f006:**
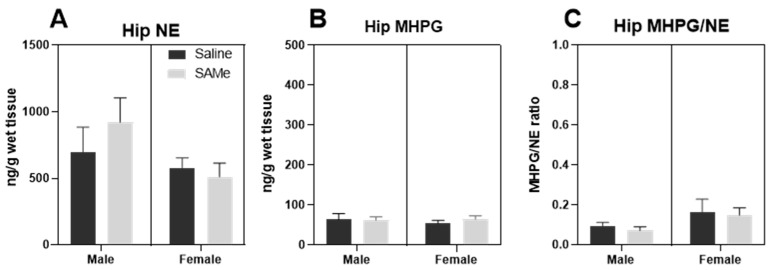
SAMe effect on norepinephrine turnover. NE (**A**), MHPG levels (**B**) and MHPG to NE ratio (**C**) in the PFC and hippocampus of Sub mice treated with SAMe did not deviate from control mice. The data are presented as mean ± SEM. Levels of NE and MHPG are presented in ng/mg, n = 6 for each group.

**Figure 7 ijms-24-10721-f007:**
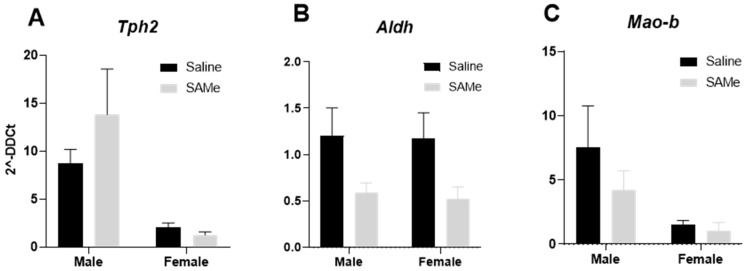
*Tph2*, *Aldh* and *Mao-b* genes expression in the hippocampus. No effect of SAMe treatment was observed for the expression of the *Tph2* (**A**), *Aldh* (**B**) and *Mao-b* (**C**) evaluated genes. The data are presented as means ± SEM. n = 6 for each group.

**Figure 8 ijms-24-10721-f008:**
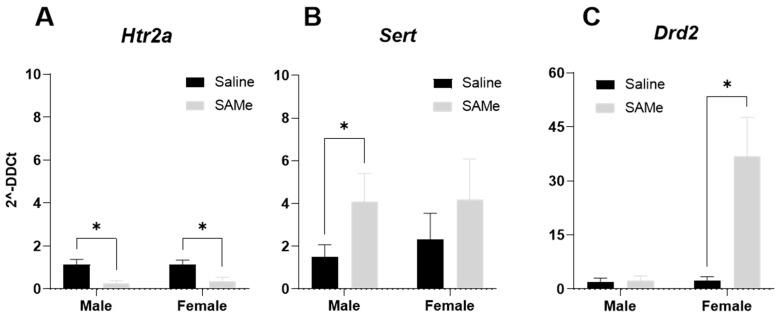
SAMe effect on *Htr2a*, *Sert* and *Drd2* genes expression in the hippocampus. (**A**) The expression levels of *Htr2a* were significantly decreased in both male and female hippocampi by SAMe; (**B**) SAMe treatment significantly increased the *Sert* gene expression only in male hippocampi, demonstrating the sex differences in response to treatment; (**C**) *Drd2* expression levels were changed in a sex-depended manner by SAMe, as a dramatic elevation in gene expression was measured in female hippocampi compared to controls. The data are presented as means ± SEM., * *p* < 0.05, n = 6 for each group.

**Figure 9 ijms-24-10721-f009:**
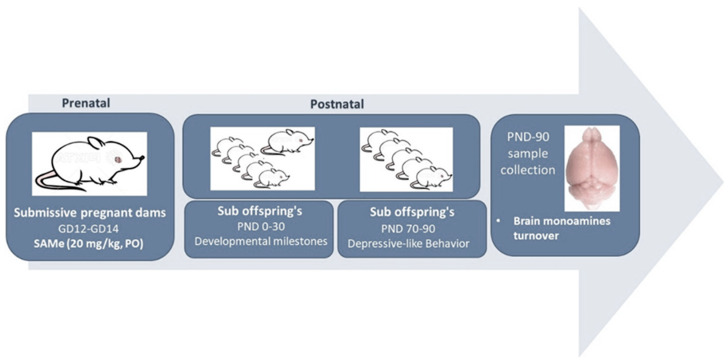
Chart flow demonstrating the flow of the experimental design.

**Table 1 ijms-24-10721-t001:** Gender-related SAMe-induced monoamine changes in PFC and hippocampi of Sub mice.

	Prefrontal Cortex		Hippocampus	
Monoamines	Male	Female	Male	Female
Serotonin	no change	increase ↑	increase ↑	increase ↑
5 HIAA	no change	no change	no change	no change
DA	increase ↑	increase ↑	no change	no change
DOPAC	increase ↑	increase ↑	no change	increase ↑
HVA	no change	increase ↑	no change	no change
Norepinephrine	no change	no change	no change	no change
MHPG	no change	no change	no change	no change

**Table 2 ijms-24-10721-t002:** Gender-related SAMe-induced changes in gene expression in PFC and hippocampi of Sub mice.

	Prefrontal Cortex		Hippocampus	
Genes	Male	Female	Male	Female
*Tph2*	no change	no change	no change	no change
*Htr2a*	Downregulated ↓	no change	Downregulated ↓	Downregulated ↓
*Sert*	no change	no change	Upregulated ↑	Trend to increase
*Mao-b*	Downregulated ↓	Trend to decrease ↓	no change	no change
*Drd2*	no change	no change	no change	Upregulated ↑

**Table 3 ijms-24-10721-t003:** List of primers used in RT-PCR.

Oligo Name	Sequence 5′ to 3′	
Glyceraldehyde 3-phosphate dehydrogenase—*Gapdh*	F	GGGGCTCTCTGCTCCTCCCTGT
	R	TGACCCTTTTGGCCCCACCCT
Serotonin receptor—*Htr2a*	F	CCTGATGTCACTTGCCATAGCTG
	R	CAGGTAAATCCAGACGGCACAG
Tryptophan hydroxylase 2—*Tph2*	F	GTTGATGCTGCGGCTCAGATCT
	R	GAAGCTCGTCATGCAGTTCACC
Serotonin Transporter—*Sert*	F	TCGCCTCCTACTACAACACC
	R	ATGTTGTCCTGGGCGAAGTA
Aldehyde dehydrogenase—*Aldh*	F	ACACAGAAGTGAAGACGGTTACTG
	R	TCTTTGGAGGCGCAGTGTGT
Monoamine oxidase B—*Mao-b*	F	GGGGGCGGCATCTCAGGTAT
	R	TGCTTCCAGAACCACCACACT
Dopamine receptor D2—*Drd2*	F	TTCCCAGTGAACAGGCGGAGA
	R	TTTGGCAGGACTGTCAGGGTT

## Data Availability

Not applicable.

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
