# Peer review of "Prenatal SAMe Treatment Changes via Epigenetic Mechanism/s USVs in Young Mice and Hippocampal Monoamines Turnover at Adulthood in a Mouse Model of Social Hierarchy and Depression"

_ijms, 2023, doi:10.3390/ijms241310721_

Round 1
Reviewer 1 Report
The paper "Prenatal SAMe treatment changes via epigenetic mechanism/s USVs in young mice and hippocampal monoamines turnover at adulthood in a mouse model of social hierarchy and depression." by Becker and colleagues is a thorough quantitative investigation of the role of S-40 adenosylmethionine (SAMe) across mouse hippocampus development. The paper is well written, constituting a good piece in the field of molecular physiology, with stated psychological and sociological repercussions. I have however some points to raise, that are currently lowering the quality of the manuscript.
Major:
- Figure 1C is barely visible. Would it be possible to access a higher resolution version of this image? I didn't find a like to HD figures in the reviewer page, and I would like to inspect the sonograms before giving a final verdict to the paper.
- Figure 2 is not as intuitive as it could be. One solution is that legends should be sorted by percentage, with the most frequent call types at the front. It is more important for the Figure to convey the results, rather than having the save class order. An even better solution is to drop entirely the bar charts, and plot these results as 4 bar plots, sorted in a decreasing order (more frequent calls to the left), with percentages clearly indicated at the top of each bar, and a shared color legend.
- The Figure around line 400 has no legend, and no reference. I would suggest giving it a proper figure number and legend, as it is important to the study design. Also in this respect, please add the sample size to the figure.
- Why only Gapdh was used? There is increasing evidence that Gapdh is not as stable and "housekeeping" as thought. The authors should show that Gapdh is as constant as at least another common housekeeping gene (e.g. Actb) across hippocampal development.
- Line 519: the authors performed Bonferroni correction, but then they report as significance P<0.05. Do they mean FDR (or P-adjusted, or corrected P)<0.05? Please clarify this in the text.
- The authors clearly used R to perform some fundamental steps of data processing (as stated on line 536), but they neither mention this in the methods/results, nor they cite it (for example, there is a good recent review about R at PMID: 35629316).
Minor:
- Please extend acronyms at their first occurrence ("SAMe", line 18; "RT-PCR", line 107).
- "Implicate" seems like the wrong verb at line 136. Rather, the authors "used", "employed" or "applied" the VocalMat algorithm.
- Typo: "previPusly", line 101.
- Typo: extra comma after "To investigate", line 135.
- Typo: "writHing", line 536
I have highlighted some minor typos and misuses of English, which I reported to the authors. A re-read of the whole text is advisable.
Author Response
We are thankful to the reviewer for the instructive comments.
The paper "Prenatal SAMe treatment changes via epigenetic mechanism/s USVs in young mice and hippocampal monoamines turnover at adulthood in a mouse model of social hierarchy and depression." by Becker and colleagues is a thorough quantitative investigation of the role of S-40 adenosylmethionine (SAMe) across mouse hippocampus development. The paper is well written, constituting a good piece in the field of molecular physiology, with stated psychological and sociological repercussions. I have however some points to raise, that are currently lowering the quality of the manuscript.
Major:
Figure 1C is barely visible. Would it be possible to access a higher resolution version of this image? I didn't find a like to HD figures in the reviewer page, and I would like to inspect the sonograms before giving a final verdict to the paper.
We replaced figure 1C with a better figure. In addition, we attached supplementary files with representative sonograms translated to be heard by the human ear.
Figure 1. The mean of call numbers differs between the groups. A – bar charts representing the average number of calls emitted by pups on postnatal days 4 and 8, respectively. Bars reflect mean values, error bars represent SEM. Two-Way ANOVA, **- p-value of multiple comparisons <0.01. B – representative oscillogram of the three-seconds sample of the USVs recording. USV oscillogram of SAMe-treated offspring (blue trace) overlaid with Saline-treated ones (orange trace) reflecting similarity in the emission intensity between groups reflecting a higher number of the calls during the certain time. C – representative sonograms obtained from sample recording of a postnatal day 4 SAMe (left panel) or Saline (right panel) -treated offspring reflecting differences in emission frequency in the treatment groups, with the SAMe-treated pups emitting vocalizations much rarely than Saline-treated ones.
- Figure 2 is not as intuitive as it could be. One solution is that legends should be sorted by percentage, with the most frequent call types at the front. It is more important for the Figure to convey the results, rather than having the save class order. An even better solution is to drop entirely the bar charts, and plot these results as 4 bar plots, sorted in a decreasing order (more frequent calls to the left), with percentages clearly indicated at the top of each bar, and a shared color legend.
We are thankful to the reviewer for this constructive comment. In the new figure 2 we plotted the mean percentage distribution of calls type sorted in a decreasing order in the sonograms of the experimental groups. Please see below.
Figure 2. Call types distribution between experimental groups. The charts reflect the percentage distribution of the different call types within the treatment groups and ages sorted in a decreasing order. A – Saline-treated offspring calls distribution at postnatal day 4. B – SAMe-treated pups calls distribution at PND4. C – Saline-treated pups calls at PND8. D – distribution of the calls within the SAMe group at PND8.
- The Figure around line 400 has no legend, and no reference. I would suggest giving it a proper figure number and legend, as it is important to the study design. Also in this respect, please add the sample size to the figure.
We added the figure number and title:
Figure 9. The chart flow demonstrates the flow of experimental design.
- Why only Gapdh was used? There is increasing evidence that Gapdh is not as stable and "housekeeping" as thought. The authors should show that Gapdh is as constant as at least another common housekeeping gene (e.g. Actb) across hippocampal development.
We agree with the reviewer that Gapdh is not an ideal and stable "housekeeping" gene for relative RT-PCR analysis for gene expression. However, it is still one of the most commonly used reference genes in research. We decided to use Gapdh in our study and measured the expression of the genes involved in the monoamine’s hippocampus turnover in relation to the GAPDH and compared the tissue of Saline treated offspring against SAMe-treated one. In our previous studies in SAMe - depression model, GAPDH was also used as reference genes- therefore we just used GAPDH to be consistent. Ref. 16 and 20 in manuscript:
- Becker, M.; Abaev, K.; Pinhasov, A.; Ornoy, A. S-Adenosyl-Methionine alleviates sociability aversion and reduces changes in gene expression in a mouse model of social hierarchy. Behavioural Brain Research 2022, 427, 113866, doi:https://doi.org/10.1016/j.bbr.2022.113866.
- Becker, M.; Abaev, K.; Shmerkin, E.; Weinstein-Fudim, L.; Pinhasov, A.; Ornoy, A. Prenatal SAMe Treatment Induces Changes in Brain Monoamines and in the Expression of Genes Related to Monoamine Metabolism in a Mouse Model of Social Hierarchy and Depression, Probably via an Epigenetic Mechanism. Int J Mol Sci 2022, 23, doi:10.3390/ijms231911898.
- Line 519: the authors performed Bonferroni correction, but then they report as significance P<0.05. Do they mean FDR (or P-adjusted, or corrected P)<0.05? Please clarify this in the text.
We corrected the text:
For statistical significance of monoamines content in the HPC in relation to gender, we performed analysis of way variance (two-way ANOVA), followed by post hoc Bonferroni test or multiple paired t-test comparison analysis [57]. For all analyses, adjusted P≤0.05 was considered significant. GraphPad Prism 9.0 software was used for Graphs preparation and statistical analysis.
- The authors clearly used R to perform some fundamental steps of data processing (as stated on line 536), but they neither mention this in the methods/results, nor they cite it (for example, there is a good recent review about R at PMID: 35629316).
Done. R package, which was used for analysis as well as macros and libraries are cited at lines 470-477 in the Ultrasonic vocalizatin.
Herein the change:
Statistical analysis was performed using platform RStudio 4.2.1 (Posit, USA, RStudio Team http://www.rstudio.com/.) utilizing the following packages: Seewave [67], tuneR [68] , ggplot2 [69], viridis [70], grid [71] and gridExtra [72] and custom-written macros. Spectogram plotting was done using Audacity® sound processing software [73]. An oscillogram plotting was done using MATLAB and Signal Analyzer Toolbox release R2023a (The MathWorks, Inc., Natick,USA). Proceeded files were subjected to custom-written macros for R to extract the numerical values of the characteristics of interest.
Minor:
- Please extend acronyms at their first occurrence ("SAMe", line 18; "RT-PCR", line 107).
Done
- "Implicate" seems like the wrong verb at line 136. Rather, the authors "used", "employed" or "applied" the VocalMat algorithm.
Done: changed to used
- Typo: "previPusly", line 101.
Done
- Typo: extra comma after "To investigate", line 135.
Done
- Typo: "writHing", line 536
Done
Comments on the Quality of English Language. I have highlighted some minor typos and misuses of English, which I reported to the authors. A re-read of the whole text is advisable.
We corrected all typos and a few more. Thanks.

Reviewer 2 Report
In this manuscript, Prenatal administration of SAMe in Sub mice with depressive-like behavior resulted in changes in hippocampal monoamines, related gene activity, and stress-induced vocalizations in the pups, indicating potential effectiveness of SAMe through an epigenetic mechanism. The manuscript is well written, but the authors must address some concerns before it is considered for publication.
1. Please enhance the clarity of Figure 1C, as the current rendition is not sufficiently legible. There are some issues (clarity, legends, and references) with the figures in this manuscript. For example, Figure 1C-make it clearer.
2. There are some typos in this manuscript such as Line 101-previously and Line 536-writing.
3. I suggest the authors read the manuscript carefully again to avoid mistakes such as typos, extra spaces, punctuations, etc.
4. For the Conclusions part, please either delete it or make it complete.
Author Response
We are thankful to the reviewer for the instructive comments.
In this manuscript, Prenatal administration of SAMe in Sub mice with depressive-like behavior resulted in changes in hippocampal monoamines, related gene activity, and stress-induced vocalizations in the pups, indicating potential effectiveness of SAMe through an epigenetic mechanism. The manuscript is well written, but the authors must address some concerns before it is considered for publication.
- Please enhance the clarity of Figure 1C, as the current rendition is not sufficiently legible. There are some issues (clarity, legends, and references) with the figures in this manuscript. For example, Figure 1C-make it clearer.
The figure was changed with a better one. In addition, we attached supplementary files with representative sonograms translated to be heard by the human ear.
Figure 1. The mean of call numbers differs between the groups. A – bar charts representing the average number of calls emitted by pups on postnatal days 4 and 8, respectively. Bars reflect mean values, error bars represent SEM. Two-Way ANOVA, **- p-value of multiple comparisons <0.01. B – representative oscillogram of the three-seconds sample of the USVs recording. USV oscillogram of SAMe-treated offspring (blue trace) overlaid with Saline-treated ones (orange trace) reflecting similarity in the emission intensity between groups reflecting a higher number of the calls during the certain time. C – representative sonograms obtained from sample recording of a postnatal day 4 SAMe (left panel) or Saline (right panel) -treated offspring reflecting differences in emission frequency in the treatment groups, with the SAMe-treated pups emitting vocalizations much rarely than Saline-treated ones.
- There are some typos in this manuscript such as Line 101-previously and Line 536-writing.
The typos were corrected
- I suggest the authors read the manuscript carefully again to avoid mistakes such as typos, extra spaces, punctuations, etc.
Done
- For the Conclusions part, please either delete it or make it complete.
We deleted the sentence that was erroneously typed from the instructions to authors

Reviewer 3 Report
Oxidative stress and neuroinflammation play crucial roles in aging. S-adenosylmethionine (SAM), a popular supplement, is a potential antioxidant and candidate therapy for depression. Recent research show that cognitive impairment and depression-like behaviors were reversed by SAM. SAM reduced neuronal cell loss, increased brain-derived neurotrophic factor level in the hippocampus, inhibited amyloid-β level and microglia activation, as well as pro-inflammatory factors levels in the hippocampus and serum. Further, SAM enhanced antioxidant capacity and attenuated cholinergic damage by reducing malondialdehyde levels, increasing acetylcholine levels, expression levels of α7 nicotinic acetylcholine receptor (α7nAChR), nuclear factor erythrocyte 2-related factor 2 (Nrf2) and heme oxygenase 1 (HO-1) in the hippocampus. Above all, SAM has a potential neuroprotective effect on ameliorating cognitive impairment in brain aging, which is related to inhibition of oxidative stress and neuroinflammation. Moreover, recent evidence indicates that SAMe increased total protein S-glutathionylation and the level of γ-glutamylcysteine ligase (GCLC). In general, in numerous experimental models, redox active compunds via hormetic dose responses display endpoints of biomedical and clinical relevance, which demonstrate to be endowed with powerful anti-inflammatory effects. Thus, interplay and coordination of redox interactions and their interaction with endogenous and exogenous antioxidant defence systems is an emerging area of reserach interest in anti-inflammatory anti-degenerative therapeutics.
This reviewer is satisfied with the significance of this study, the care in which the study was performed, and the implications of the results for human health. Results presented are interesting and the questions posed are of extremely high interest, However, the paper does not give adequate definitive information. Pending major points, this paper can be accepted
Minor concerns:
1. Authors should measure and ascertain in their expermental settings the role of Nrf-2 and Nrf-2 dependent vitagenes (HO-1 and gamma-lyase, as an example) and establish proper correlations between redox state and behavioural effects of SAM.
2. Given the relationship between vitagene network and its possible biological relevance in the defense mechanisms against oxidative stress-driven degenerative diseases, Authors can mention in the discussion appropriately this aspect (See and quote please Calabrese V., et al., (2009) Biofactors 35, 146-160; ; Calabrese et al., 2010, Antiox. Redox Signal 13,1763; Calabrese et al., Nature Neurosci., 2007 8, 766; Siracusa R, et al., 2020, 9(9):824. doi: 10.3390/antiox9090824).
Minor concerns:
1. A uthors should measure in their expermental settings the role of Nrf-2 dependent vitagenes (HO-1 and gamma-lyase) and establish proper correlations between redox state and behavioural effects of SAM.
2. Given the relationship between vitagene network and its possible biological relevance in the defense mechanisms against oxidative stress-driven degenerative diseases, Authors can mention in the discussion appropriately this aspect (See and quote please Calabrese V., et al., (2009) Biofactors 35, 146-160; ; Calabrese et al., 2010, Antiox. Redox Signal 13,1763; Calabrese et al., Nature Neurosci., 2007 8, 766; Siracusa R, et al., 2020, 9(9):824. doi: 10.3390/antiox9090824).
Author Response
We are thankful to the reviewer for the instructive comments.
Oxidative stress and neuroinflammation play crucial roles in aging. S-adenosylmethionine (SAM), a popular supplement, is a potential antioxidant and candidate therapy for depression. Recent research show that cognitive impairment and depression-like behaviors were reversed by SAM. SAM reduced neuronal cell loss, increased brain-derived neurotrophic factor level in the hippocampus, inhibited amyloid-β level and microglia activation, as well as pro-inflammatory factors levels in the hippocampus and serum. Further, SAM enhanced antioxidant capacity and attenuated cholinergic damage by reducing malondialdehyde levels, increasing acetylcholine levels, expression levels of α7 nicotinic acetylcholine receptor (α7nAChR), nuclear factor erythrocyte 2-related factor 2 (Nrf2) and heme oxygenase 1 (HO-1) in the hippocampus. Above all, SAM has a potential neuroprotective effect on ameliorating cognitive impairment in brain aging, which is related to inhibition of oxidative stress and neuroinflammation. Moreover, recent evidence indicates that SAMe increased total protein S-glutathionylation and the level of γ-glutamylcysteine ligase (GCLC). In general, in numerous experimental models, redox active compunds via hormetic dose responses display endpoints of biomedical and clinical relevance, which demonstrate to be endowed with powerful anti-inflammatory effects. Thus, interplay and coordination of redox interactions and their interaction with endogenous and exogenous antioxidant defence systems is an emerging area of reserach interest in anti-inflammatory anti-degenerative therapeutics.
This reviewer is satisfied with the significance of this study, the care in which the study was performed, and the implications of the results for human health. Results presented are interesting and the questions posed are of extremely high interest, However, the paper does not give adequate definitive information. Pending major points, this paper can be accepted
Minor concerns:
- Authors should measure and ascertain in their expermental settings the role of Nrf-2 and Nrf-2 dependent vitagenes (HO-1 and gamma-lyase, as an example) and establish proper correlations between redox state and behavioural effects of SAM.
We are thankful to the reviewer for the constructive comments. Indeed, we are interested to investigate the correlation between redox potential in brain tissue and the behavioral effects of SAMe treated offspring. We planned our future results to investigate this issue in the next round of the study and obtain sufficient biological material. Unfortunately, we do not have sufficient material to perform the suggested experiments. In our previous experiments on the role of SAMe ameliorating the ASD like behavior in a mouse model of VPA – induced ASD like behavior, SAMe indeed reduced the oxidative stress found in the prefrontal cortex of VPA - induced ASD like behavior (Ref 63: Ornoy, A.; Weinstein-Fudim, L.; Tfilin, M.; Ergaz, Z.; Yanai, J.; Szyf, M.; Turgeman, G. S-adenosyl methionine prevents ASD like behaviors triggered by early postnatal valproic acid exposure in very young mice. Neurotoxicology and Teratology 2019, 71, 64-74, doi:https://doi.org/10.1016/j.ntt.2018.01.005.)
- Given the relationship between vitagene network and its possible biological relevance in the defense mechanisms against oxidative stress-driven degenerative diseases, Authors can mention in the discussion appropriately this aspect (See and quote please Calabrese V., et al., (2009) Biofactors 35, 146-160; ; Calabrese et al., 2010, Antiox. Redox Signal 13,1763; Calabrese et al., Nature Neurosci., 2007 8, 766; Siracusa R, et al., 2020, 9(9):824. doi: 10.3390/antiox9090824).
Thanks for these comments. We added to the discussion the following paragraphs:
The brain is very susceptible to the damage of oxidative stress as well as nitrosative stress. Calabrese et al have shown that physiological concentrations of NO are involved in a variety of regulatory mechanisms in the brain, while high doses are neurotoxic [62]. SAMe, in addition to its role in DNA methylation, may also serve as an important antioxidant [3]. Hence, it is possible that in our study SAMe normalized some of the biochemical and molecular changes observed in the untreated Sub mice via its antioxidant capacity. Indeed, in our previous studies on a mouse model of Autistic like behavior induced by early postnatal treatment with VPA, we found that concomitant treatment with SAMe alleviated not only the autistic like behavior but also the oxidative stress induced by VPA as observed in the prefrontal cortex at 60 days of age [1].
In addition, in a rodent model of-galactose-induced brain aging Zhang et al [2] have shown that chronic treatment with SAMe enhanced antioxidant capacity and reduced the expression levels of α7 nicotinic acetylcholine receptor (α7nAChR), nuclear factor erythrocyte 2-related factor 2 (Nrf2) and heme oxygenase 1 (HO-1) genes in the hippocampus and improved cognitive impairment in rat behavior [2].
It was shown that rapid degradation of 5-HT, NE and DA by MAO enzyme leads to the formation of a potentially neurotoxic species, such as hydrogen peroxide and ammonia that resulted in the production of reactive oxygen species (ROS) inducing mitochondrial damage and neuronal apoptosis (reviewed in [3]). We assume that SAMe inhibited the degradation of monoamines which, in turn, may diminish the formation of ROS and NOS. However, we did not measure in our present study the redox potential to elucidate possible SAMe effect on redox homeostasis.
- Ornoy, A.; Weinstein-Fudim, L.; Tfilin, M.; Ergaz, Z.; Yanai, J.; Szyf, M.; Turgeman, G. S-adenosyl methionine prevents ASD like behaviors triggered by early postnatal valproic acid exposure in very young mice. Neurotoxicology and Teratology 2019, 71, 64-74, doi:https://doi.org/10.1016/j.ntt.2018.01.005.
- Zhang, Y.; Ma, R.; Deng, Q.; Wang, W.; Cao, C.; Yu, C.; Li, S.; Shi, L.; Tian, J. S-adenosylmethionine improves cognitive impairment in D-galactose-induced brain aging by inhibiting oxidative stress and neuroinflammation. Journal of Chemical Neuroanatomy 2023, 128, 102232, doi:https://doi.org/10.1016/j.jchemneu.2023.102232.
- Bortolato, M.; Chen, K.; Shih, J.C. Monoamine oxidase inactivation: From pathophysiology to therapeutics. Advanced drug delivery reviews 2008, 60, 1527-1533, doi:https://doi.org/10.1016/j.addr.2008.06.002.
- Ornoy, A.; Weinstein-Fudim, L.; Tfilin, M.; Ergaz, Z.; Yanai, J.; Szyf, M.; Turgeman, G. Sadenosyl methionine prevents ASD like behaviors triggered by early postnatal valproic acid
exposure in very young mice. Neurotoxicology and Teratology 2019, 71, 64-74,
doi:https://doi.org/10.1016/j.ntt.2018.01.005. - Zhang, Y.; Ma, R.; Deng, Q.; Wang, W.; Cao, C.; Yu, C.; Li, S.; Shi, L.; Tian, J. Sadenosylmethionine improves cognitive impairment in D-galactose-induced brain aging by
inhibiting oxidative stress and neuroinflammation. Journal of Chemical Neuroanatomy 2023,
128, 102232, doi:https://doi.org/10.1016/j.jchemneu.2023.102232. - Bortolato, M.; Chen, K.; Shih, J.C. Monoamine oxidase inactivation: From pathophysiology to
therapeutics. Advanced drug delivery reviews 2008, 60, 1527-1533,
doi:https://doi.org/10.1016/j.addr.2008.06.002.

Round 2
Reviewer 1 Report
I thank the authors for taking a lot of time and effort into improving their manuscript. I particularly appreciated the inclusion of the audio sonograms as supplementary files. I believe the paper to be ready for publication.
Reviewer 3 Report
Accept